# An Evaluation of Orthotics on In-Toeing or Out-Toeing Gait

**DOI:** 10.3390/healthcare13050531

**Published:** 2025-02-28

**Authors:** Harshavardhan Bollepalli, Carter J. K. White, Jacob Dane Kodra, Xue-Cheng Liu

**Affiliations:** 1Department of Orthopedic Surgery, Medical College of Wisconsin, Milwaukee, WI 53226, USA; hbollepalli@mcw.edu (H.B.); cawhite@mcw.edu (C.J.K.W.); jkodra@mcw.edu (J.D.K.); 2Department of Orthopedic Surgery, Children’s Wisconsin, Medical College of Wisconsin, Greenfield, WI 53227, USA

**Keywords:** orthotics, rotational deformities, gait analysis, neuromuscular conditions

## Abstract

**Background and Objectives**: In-toeing and out-toeing gait are rotational deformities commonly observed in children with neuromuscular conditions. These gait abnormalities often result from internal tibial torsion, increased femoral anteversion, and metatarsus adductus. This study was conducted to create a comprehensive evaluation of the effectiveness of lower extremity orthotics as a non-operative treatment option, given their regular use in clinical settings. The aim of this literature review was to understand the efficacy of various orthotic devices in correcting rotational deformities in the transverse plane, thereby improving ambulation stability and 3D joint motion. **Materials and Methods**: Literature published after 1 January 1990 was reviewed, utilizing databases such as CENTRAL (Wiley), CINAHL (EBSCO), Medline (OVID), Scopus (Elsevier), and Web of Science (Clarivate). In totality, 13 studies were included, evaluating 365 participants with neuromuscular conditions using various orthotic devices. **Results**: Among these studies, two were randomized control trials (Level 1), nine were quasi-experimental studies (Level 2), and two were case studies (Level 4). Quality assessment determined that 69% of the included studies had a low risk of bias, while 31% demonstrated a moderate risk. Compression garments and rotational systems showcased the greatest change in proximal lower extremity rotation at 19.73° ± 1.57 and 24.13° ± 8.49, respectively. The most significant difference in foot progression angle is through the use of rotational systems, 19° ± 26.87. **Conclusions**: In a short-term treatment, children with neuromuscular disorders exhibiting in-toeing or out-toeing gait may benefit from different types of orthoses. Compression garments may aid joint alignment and enhance proprioception, rotational systems correct alignment with precise adjustability, AFOs that achieve effective stabilization can deliver benefits in the transverse plane, and foot orthotics may be appropriate for mild gait abnormality management.

## 1. Introduction

In-toeing and out-toeing are gait abnormalities commonly seen in pediatric populations [1,2]. They are characterized as rotational abnormalities resulting in an inappropriately rotated foot, with in-toeing being described as a unilateral or bilateral internal rotation of the longitudinal axis of the foot compared to the line of progression, as opposed to out-toeing consisting of external rotation [1,2]. These rotational deformities are clinical symptoms arising from an underlying anatomic or functional dysfunction consisting of range of movement, neurological, and static skeletal disorders [1,2]. Three primary deformities that lead to in-toeing include tibial torsion, metatarsus adductus, and femoral anteversion [3,4]. In comparison, the primary deformities in out-toeing gait include pelvic external rotation, hip external rotation, or external tibial torsion [1]. In these deformities, femoral anteversion refers to a deviated angle between the femoral neck and the femoral shaft caused by forward torsion of the femoral neck [5], whereas tibial torsion describes the twisting of the tibia either internally or externally [6,7]. Both femoral anteversion and tibial deformities may lead to increased internal rotation and muscle activation at the hip and knee [8,9,10,11]. If these deformities are left unaddressed, they may persist and worsen into adulthood, potentially leading to excessive subtalar joint rotation [12], hip joint arthritis [13], and patellofemoral injuries [13].

Some causes of in-toeing and out-toeing gait abnormalities are linked with conditions such as cerebral palsy and spina bifida [14,15,16]. These neuromuscular disorders can affect bone growth and orientation, muscle strength and tone, and movement coordination, creating an in-toeing or out-toeing gait [17,18]. These patients may also exhibit joint laxity, muscle imbalances, hip dislocations, knee or foot deformities, and are often prone to in-toeing gait patterns [19,20].

To effectively manage these abnormal rotations, the site of deformity must be accurately diagnosed by assessing the hip’s internal and external rotation, foot progression angle (FPA) [1,21], thigh–foot angle (TFA) [22,23], foot shape, and the transmalleolar axis (TMA) [23,24] in the knee-flexed position [25]. FPA is considered as one of commonly used standard parameters to diagnose abnormal gait. This quantitative measurement system allows clinicians to quantify motion and characterize these pathologies [26]. Unlike femoral anteversion and tibial torsion, which focus on the femur and tibia specifically, FPA is a measurement of the entire lower extremity chain presenting a summative description of rotational deformities [26]. Along with long-term consequences of arthritis and lower extremity instability [12,13], these deformities can lead to increased pain in the future, as supported by a retrospective study which identified 43 out of 50 children with increased internal hip rotation reporting pain on physical examination [21]. Depending on the level of the lower extremity affected, it can impact the adjacent musculature and result in rotational forces pulling joint alignment into an internal rotation, causing in-toeing and the need to compensate with other structures [26].

Conservative treatment options such as orthotics have proven capable of correcting misalignments and abnormal gait patterns [27,28,29]. This is accomplished through providing stability of movement and support of joints or muscles, reducing the muscle spasticity and energy for ambulation, and improving the quality of life. Currently, there are limited literature reviews that have been published on comparing multiple orthotic treatments for children with neuromuscular conditions and in-toeing or out-toeing gait. There remains a lack of comprehensive reviews that evaluate multiple orthotic types together, including TheraTogs, twister cables (TCs), twister wrap orthoses (TWOs), TheraSuits, compression garments, ankle and foot orthoses (AFOs), and foot orthoses (FOs). A recent debate has focused on the role of plantar orthotics, which have shown efficacy in reducing musculoskeletal stress and improving distal weakness in the lower extremities through redistributing plantar pressure and maintaining correct alignment, allowing for improved gait stability [30,31,32]. Given the potential for long-term complications, continued debate on orthotic efficacy, and the wide range of orthotic options available, a thorough review and comparison of the efficacy of these orthotic devices is essential for effective clinical decision making and crucial to comprehensively evaluate their impact on gait abnormalities, particularly concerning transverse plane rotation. 

The goal of this study is three-fold: (1) to compare the efficacy and clinical findings of various orthotic devices in patients with in-toeing or out-toeing gait deformities; (2) to compare different orthotic influences on 3D kinematics and kinetics of the lower extremity, especially on the transverse plane movements; (3) to assess the demographic data and the quality of life for children using each orthotic.

## 2. Materials and Methods

### 2.1. Protocol and Registration

The study protocol was registered on Open Science Framework (osf.io) at https://osf.io/ecqm2/, accessed on 22 April 2024. (Registration Information: The literature review was registered to an open access platform at Open Science Framework (OSF)).

### 2.2. Search Strategy

The current review was conducted on 29 May 2024 through using the following databases to find scientific articles: CENTRAL (Wiley) (Hoboken, NJ, USA), CINAHL (EBSCO) (Ipswich, MA, USA), Medline (OVID) (Norwood, MA, USA), Scopus (Elsevier) (Amsterdam, Netherlands), and Web of Science (Clarivate) (London, United Kingdom), which is outlined in Figure 1. In Ovid Medline, we used MeSH terms and free text terms related to neuromuscular diseases, neurologic gait disorders, orthotic devices, and the pediatric population. The complete search strategy is included in Table A1 in Appendix A.

### 2.3. Eligibility Criteria

This study’s inclusion criteria focused on children with neuromuscular conditions, such as cerebral palsy and spina bifida, who present with an in-toeing or out-toeing rotational deformity. Individuals without neuromuscular conditions causing rotational deformities and aged 18 years and older were excluded. Studies that have used orthotic devices to correct lower extremity rotational deformities were chosen to be included. The inclusion criteria are based on the articles discussing one or more of the following key phrases: “in-toeing”, “out-toeing”, “neuromuscular conditions”, “transverse plane”, “rotational deformities”, “cerebral palsy”, “spina bifida”, “rehabilitation devices”, “assisted devices”, “TheraTogs”, “Twister Cables” (TC), “Tibia Counter Rotator” (TCR), “Twister Wrap Orthoses” (TWO), “compression garments”, “therasuit”, “lycra garment”, “compression leggings”, “AdeliSuit”, “Dynamic Elastomeric Fabric Orthoses” (DEFO), “Hinged Ankle Foot Orthosis” (HAFO), “Solid Ankle Foot Orthosis” (SAFO), “Supramalleolar orthosis” (SMO), “ankle foot orthosis” (AFO), and “Static Ground Reaction Ankle Foot Orthoses” (sgrAFO). Studies that have not described the listed types of orthotics, did not discuss the transverse plane or rotational deformities, or were published before January 1st, 1990, were excluded. Additionally, any unclear data relating to the variables in the inclusion criteria were not included. The comparisons of the included studies must involve different orthotic treatments or be comparisons between barefoot ambulation or baseline interventions such as physical therapy. The main outcomes of interest include changes in internal and external rotations in 3D kinematics and changes in FPA.

### 2.4. Study Screening

The article selection process began by removing duplicates, followed by title screening and eventually abstracts and full texts of the remaining studies, using the review software Rayyan (https://www.rayyan.ai/) (accessed on 29 May 2024) (Cambridge, MA, USA). For each synthesis, the intervention characteristics of the studies were tabulated and compared against the pre-defined inclusion criteria: (1) the type of orthotic intervention (TC, TWO, AFO, garment, etc.); (2) study population with gait disorder (e.g., children with in-toeing or out-toeing gait); (3) outcome measures (e.g., spatio-temporal data, foot progression angle, and 3D joint kinematics); (4) age from starting to walk to 18 years old. Studies that met these criteria were included in the synthesis. Articles were reviewed by two independent reviewers, who were blinded to each other’s decisions. Conflicts regarding inclusion or exclusion were resolved by a third reviewer. Exclusion criteria include: (1) age >18 years old; (2) post-surgery with soft tissue or bone osteotomy or spine; (3) no orthotics. A risk of bias assessment for the included studies was conducted.

### 2.5. Data Collection

One author extracted the critical data from each article to allow a comparison of the interventions. These key data included the study design, type of intervention, subject characteristics (number of participants, neuromuscular condition), spatio-temporal data, and gait analysis data such as FPA outcomes and 3D joint kinematics, focusing on the transverse plane. The changes in internal/external rotation and FPA were gathered from multiple clinical studies (Table 1, Table 2, Table 3, Table 4, Table 5, Table 6 and Table 7). The findings from each clinical study regarding changes in the transverse plane were compared and depicted for each orthotic category: compression garment [19,33,34,35,36], rotational systems [20,37], ankle and foot orthotics [38,39,40], and insole and wedges [13,41,42] (Table 6). A similar process was accomplished in comparing the average FPA between all four orthotic categories (Table 7).

### 2.6. Quality Assessment

The level of evidence was assigned to each article based on the hierarchy outlined by Burns et al. in their 2011 study [43]. Two reviewers (HB, CW) evaluated all included studies independently. Inconsistencies were shown to a third reviewer (JK) for the final decision.

The Cochrane Collaboration “risk of bias” tool was used to evaluate the included randomized control trial studies [44], which included the following: sequence generation; allocation concealment; blinding of participants and personnel; blinding of outcome assessors; incomplete outcome data; selective outcome reporting; and other biases. Each item was rated as unclear, low risk, or high risk of bias. We did not exclude any studies for methodological quality.

For the quasi-experimental studies, the Risk of Bias In Non-randomized Studies of Interventions (ROBINS-I) tool [45] was used to assess bias across the following domains: confounding; participant selection; classification of interventions; deviations from intended interventions; missing data; measurement of outcomes; and selection of the reported result. Each domain was rated as low risk, moderate risk, serious risk, critical risk, or no information to reflect the risk of bias in non-randomized studies. We did not exclude any studies for methodological quality.

Additionally, the Joanna Briggs Institute Critical Appraisal Checklist was used to assess case series and case studies [46], examining the inclusion criteria; clear description of the participants; measurement of outcomes; appropriate statistical analysis; and clear reporting of findings. Each item was assessed for potential bias in reporting and reliability.

### 2.7. Outcome Measures

Among the included clinical studies, children of ages 2 to 15 years old had various neuromuscular conditions, such as diplegic and hemiplegic cerebral palsy, spina bifida, Duchenne muscular dystrophy, and Down syndrome. The types of orthotics that were investigated include rotational systems such as twister cables, compression garments such as TheraTogs and Lycra garments, various forms of ankle and foot orthotics (AFO, HAFO, SAFO), and insole and wedges. The outcomes that were analyzed include changes in the FPA [47] and 3D joint kinematics [48], specifically focusing on the transverse plane. For each outcome across different orthotic systems, the mean differences between subject age, FPA, and 3D joint kinematics were used as effect measures in the synthesis and presentation of results.

### 2.8. Statistical Analysis

Differences among the FPA and 3D joint kinematics from the included studies were compared among the different types of orthotics. Statistical analyses for combining means and standard deviations from the individual studies were performed using R version 4.3.2 (R Foundation for Statistical Computing). To ensure the accuracy of the analysis, the original sample sizes, means, and standard deviations for each study were carefully extracted from the corresponding published reports. These data were then aggregated using the methods suggested by Higgins et al. (2019) [49], which provide robust formulas for synthesizing summary statistics across studies. Specifically, weighted means and pooled standard deviations were calculated, considering the sample sizes and variability of each study. These comparisons are shown in Table 6 and Table 7, providing a general comparison between kinematic findings.

## 3. Results

### 3.1. Search Strategy

The initial search gathered 1299 articles, of which 462 were duplicates. Then, 301 articles were examined by reviewers. Thirteen studies were included, evaluating 365 participants with neuromuscular conditions, including diplegic and hemiplegic cerebral palsy, spina bifida, myelomeningocele, Duchenne muscular dystrophy, and Down syndrome (Figure 1). All 13 studies were based on pediatric patient populations. Institutional access through Medical College of Wisconsin libraries was used to access all articles.

### 3.2. Study Quality

This literature review included 13 clinical studies. Of the 13 studies analyzed, 15% were Level 1, 69% were Level 2, 15% were Level 4. Collectively, two were randomized control trials, nine were quasi-experimental studies, five were case–control studies, and two were case studies. Regarding the risk of bias, 69% of the studies had a low risk of bias, and 31% of the studies had a moderate risk of bias.

### 3.3. Types of Orthotics and Their Impacts

The orthotics mentioned in the tables include TheraTogs [19,20,35], SMOs [38,39], high-top shoes [33], TWOs [33], other 3D supporting garments [34,36], medial-wedge insoles [41] (MWI), DEFOs [50], a custom-made TCR [37], shoe wedges [12,13,34,38,39,41,42], TCs [20], general AFOs [51,52], and specific AFOs such as sgrAFOs [19], HAFOs [38], and SAFOs [38,53]. 

The orthotics were grouped into four categories: compression garments (Table 2), rotational systems (Table 3), AFOs (Table 4), and insoles and wedges (Table 5). Table 6 depicts the changes in hip and knee rotation in the transverse plane. Compression garments show a notable improvement in hip rotation with a combined mean of 19.73 degrees and a standard deviation of 1.57. However, no data were available for statistical analysis for knee rotation using compression garments. In contrast, rotational systems showed improvements in both hip rotation with a change in mean of 8.5 degrees and knee rotation with a change in mean of 24.13, although no standard deviations were able to be reported. Additionally, Table 7 illustrates changes in FPA in the transverse plane for all four orthotic groups. Rotational systems depicted the largest FPA improvement of 19 degrees, but with high variability with a standard deviation of 26.87 degrees. Compression garments showed the smallest change in FPA of 4.86. Insoles and wedges demonstrated a moderate FPA of 13.95 degrees with a high degree of variability. Ankle and foot orthotics depicted no improvement in FPA.

## 4. Discussion

The present literature review aimed to examine the effects of different orthotic types on changes in the transverse plane among patients with neuromuscular conditions. Of the included articles, 38% (5/13) discussed the impact of compression garments on the transverse plane [19,33,34,35,36]. These studies provide a strong evidence base suggesting that compression garments, such as TheraTogs, can improve knee alignment, allowing for enhanced muscle recruitment at the hip joint in the swing phase and rotation of the leg for improved FPA. These findings complement a study by Flanagan that underscored a 76% increase in balance after TheraTogs intervention [35] and a study by Emara assessing the quality of life and functional independence with TheraTogs use [54]. Similar to findings by Degelean [34], their results showcase how improved internal and external rotation can strengthen joint stability, increase mobility, and decrease pain and fall rates. This is supported by a study by Schelhaas et al. which observed a 4.4° decrease in out-toeing with TheraTogs intervention, indicating a marked improvement in FPA [55].

Of the studies discussing compression garments, 2/5 compared compression garments with the addition of AFOs or high-top shoes [19,33]. Abd El-Kafy also highlighted no significant differences between bilateral hip and knee rotational angles between TheraTogs and “TheraTogs + static ground reaction AFOs” conditions before and after treatment. Since AFOs are understood to increase stability of the ankle in the coronal plane, this explains how significant changes in the transverse plane, affected by TheraTogs, remain the primary contributing factor to the improvement of in-toeing or out-toeing symptoms. Davoudi et al. (2022) [33] demonstrated that in children with diplegic cerebral palsy, the use of high-top shoes combined with TWO significantly reduced center pressure sway in both anterior–posterior and medial–lateral directions, thus enhancing balance and correcting transverse plane misalignment through femoral de-rotation. This correction notably improved hip internal rotation control, thus potentially enhancing motor functionality and reducing fall risk. A randomized controlled trial by Emara et al. [56] supports these findings through showing how corrected foot pressure distribution was more significant in TheraTogs use compared to conventional physical-therapy-corrected foot pressure distribution.

Other variations, such as Lycra garments, provide dispersed pressure that improves muscle activation and proprioception by applying external force vectors to counteract internal rotation at the hip, knee, and ankle levels as depicted in Rennie et al. [36]. Correspondingly, Kim et al. [50] found an 8–9° improvement in hip rotation in the transverse plane, while Cunha et al. [57] noted enhanced proprioceptive feedback through pressure receptors, facilitating targeted muscle activation. Rennie’s study further highlighted how Lycra garments lacked sufficient foot and ankle support, which may limit overall joint stability. These findings suggest the importance of proper compression garment wrapping of the entire lower extremity to improve stabilization and mitigate internal and external forces. Overall, due to the versatility, muscle activation, and proprioceptive features of compression garments, they can be an optimal option for individuals with various neuromuscular conditions. The findings of the current review align with the previous studies regarding the rotational improvement at the hip joint and FPA improvement.

Fifteen percent of the included articles (2/13) discussed the impact of rotational systems on the transverse plane [20,37]. The results of these articles indicate that TCs and TCRs can be a valuable aid for children with neuromuscular disorders that result in severe traversal rotational deformities at multiple levels. Unlike compression garments, TCs and TCRs provide rigid, controllable rotational forces that enforce corrective alignment of the lower extremities [20], which may provide extensive benefit as the patient develops due to changes in musculoskeletal structures and joint forces. TCs consist of cables and straps that attach to a brace or shoe and provide correction at the femur and tibia. In contrast, the TCR acts as a splint, providing isolated rotational correction at the knee joint and tibia [20]. Both orthotic devices showed statistical significance in improving internal rotation. TCs have demonstrated improved hip rotation by providing external changes of 23° on the left lower extremity and 25° on the right lower extremity, along with enhanced external rotation in the FPA with 30° on the left lower extremity and 13° on the right lower extremity [20]. In comparison, the TCR was shown to be effective as an initial treatment option for internal tibial torsion [37]. Compared to TheraTogs, TCs had decreased improvements with hip external rotation but greater overall FPA improvement [20]. Our analysis reported that rotational systems have a greater impact on reducing the FPA compared to compression garments (See Table 7). TheraTogs may be more beneficial for patients with hip internal rotation, as the external forces at the hip can decrease knee stress and prevent future complications that may include knee arthritis and ligament instability [20]. Alternatively, TCs or TCRs may be better indicated for patients who require precise correction at multiple joints including the hip and knee rather than the pelvis [20]. Similarly, our analysis demonstrates how rotational systems influenced both hip and knee levels of the lower extremity, as compared to compression garments that only influenced the hip (see Table 6).

Although TCs can provide rotational changes at multiple joints due to the structure and attachment points of the orthotic, a patient satisfaction survey exhibited a preference for TheraTogs over TCs [20,36]. This partiality may be attributed to the considerable size of TCs compared to TheraTogs, which may impact the child’s willingness to wear the orthotic for long periods of time [20]. While the TCR provides a mild improvement in the transverse plane of the knee with a 5° improvement in the right leg and 0° in the left leg, unlike TCs, the TCR allows for independent movement of both legs [37]. TCRs may be a more suitable treatment for patients with tibial torsion who cannot wear TCs for long periods. However, findings in Richards et al. are similar to those in Rennie et al. as parents in the study highlighted their children’s challenges with using the restroom and chafing near joint lines [36]. Even with Rennie et al.’s study showing a moderate risk of bias, these investigations show that both garments and rotational systems can prove uncomfortable and demonstrate minimal effectiveness if the individual is unwilling to use them consistently. 

Twenty-three percent of the included articles (3/13) discussed the impact of AFOs on the transverse plane [38,39,40]. The results of these articles indicate conflicting findings surrounding the utility of this class of orthotics. One study found AFOs introduced further internal rotation by 7 degrees [40]. Some findings suggests that tibial torsion shoes with either AFOs or SMOs failed to demonstrate significant correlations with gait parameters [39]. Other literature on these orthotics has demonstrated increased and varied transverse motion, such as at the knee joint [52] and increased external tibial torsion, leading to decreased knee extension mobility [53,58,59]. The sgrAFO showed no statistically significant differences in bilateral hip and knee angles between groups treated with TheraTogs versus “TheraTogs and sgrAFO” [19]. It supports the limited benefit of AFOs for pathologies affecting the transverse plane [19]. Both Carmick et al. [38] and Fatone et al. [58] observed the utility of AFOs, including SAFOs, HAFOs, and SMOs, highlighting the importance of proper alignment of the mechanical joint axis. Specifically, Carmick et al. discusses the need for a neutral subtalar joint (STJ) position for beneficial ambulation outcomes. They argue that having a neutral STJ will lead to proper biomechanical alignment of the hip, knee, and ankle, leading to reduced gait abnormalities. As compared to both compression garments and rotational systems, our study shows that AFOs have no impact on FPA (see Table 7).

Twenty-three percent of the included articles (3/13) discussed the impact of foot orthotics on the transverse plane [13,41,42]. The results of all three studies show significant improvement in FPA. For example, Munuera et al. [13] found a 5.30° improvement in the FPA in patients with in-toeing. However, there is conflicting evidence on the impact of insoles and wedges on femoral anteversion. A study by Ganjehie et al. found that gait plate insoles increased FPA between foot longitudinal axis and the center of pressure in the anterior–posterior direction in children with femoral anteversion [12]. Contrastingly, a randomized control trial by Parian et al. found that children suffering from excessive femoral anteversion experienced no statistically significant difference in FPA during the first four weeks of using the orthotic [42]. Due to confounding variables, these differences may be attributed to the time of usage for the orthotic.

Compliance and comfortability are major factors for clinicians to take into account when prescribing orthotics. The prescribed orthotic may be beneficial in correcting malalignment; however, if it consistently causes discomfort and the child becomes averse to treatment compliance, then the efficacy of the orthotic will be limited. A parental satisfaction survey in Richards et al. [20] found that parents reported higher satisfaction with TheraTogs in regard to comfortability (rated 1 for strongly agree), improved balance (rated 1 for strongly agree), and child’s willing to wear the garment all day in comparison to twister cables where parents rated a 5 for strongly disagree. Although twister cables may be more effective and accurate in correcting lower extremity malalignment, without potential risks such as overcorrection, the bulkiness and weight of the orthotic may limit in treatment success, especially in younger children, regarding the compliance of the patient [60,61]. A potential method to mitigate non-compliance may to be design custom-fit orthotics that limit discomfort observed in standard models [62].

In the current literature review, there are a few limitations that should be acknowledged. The heterogeneity among the included studies due to the differences in study design, evaluated orthotic, and outcomes places restrictions on performing a meta-analysis or statistical comparison between interventions. There are a limited number of studies discussing orthotics such as those in the rotational system category, limiting the data for a comprehensive comparison. Additionally, variations in patient demographics, ages, and number of participants in the included articles impact the conclusions that were drawn. The limited long-term follow-up in a majority of the studies places restrictions on drawing conclusions on the efficiency and sustainability of the orthotic intervention in transverse plane correction. Regarding the review process, reliance on published literature may have created a publication bias, as studies with significant findings are more often published. Thus, these studies may overrepresent the efficacy of these orthotic interventions in lower extremity malalignment, possibly limiting the generalizability of this study. To mitigate this for future studies, authors may conduct a systematic review on relevant literature, broaden their inclusion criteria with a greater number of orthotics, and include outcome measures, studies before 1990, or additional databases to overcome this limitation. Table 6 and Table 7 are representations of listed findings of rotational changes in the transverse plane and do not portray statistical significance.

## 5. Conclusions

Orthotic treatments vary widely but have a notable capacity to remediate gait abnormalities in the transverse plane and in pediatric patients suffering from neuromuscular conditions. While this study primarily focused on in-toeing and out-toeing gait, children with neuromuscular disorders often have complicated 3D movement abnormalities and different functional requirements at age-related milestones. Clinically, optimal orthotic selection is hardly indicated for sole management of the transversal plane. Comprehensive physical examination and gait analysis tools for assessing rotational deformities are imperative for physicians to provide effective orthotic treatment in these patients. This review included 13 clinical studies with the majority characterized by Level 2 evidence and offering primarily moderate certainty in conclusions. While 69% of studies had a low risk of bias, 31% were at moderate risk. Notably, 92% of included studies (12/13) demonstrated that orthoses, such as compression garments, rotational systems, as well as insoles and wedges, each improved internal or external rotation in the transverse plane. Compression garments provide benefit through joint alignment correction and proprioception enhancement. Rotational systems provide similar advantages, with the added capability of adjustability to specific degrees during treatment progression, although compliance among children remains a challenge. While AFOs and insoles or wedges present mixed evidence, achieving effective stabilization is crucial for AFOs to deliver measurable benefits in the transverse plan. Foot orthotics (FOs), conversely, may be more appropriate for managing mild gait abnormalities.

## Figures and Tables

**Figure 1 healthcare-13-00531-f001:**
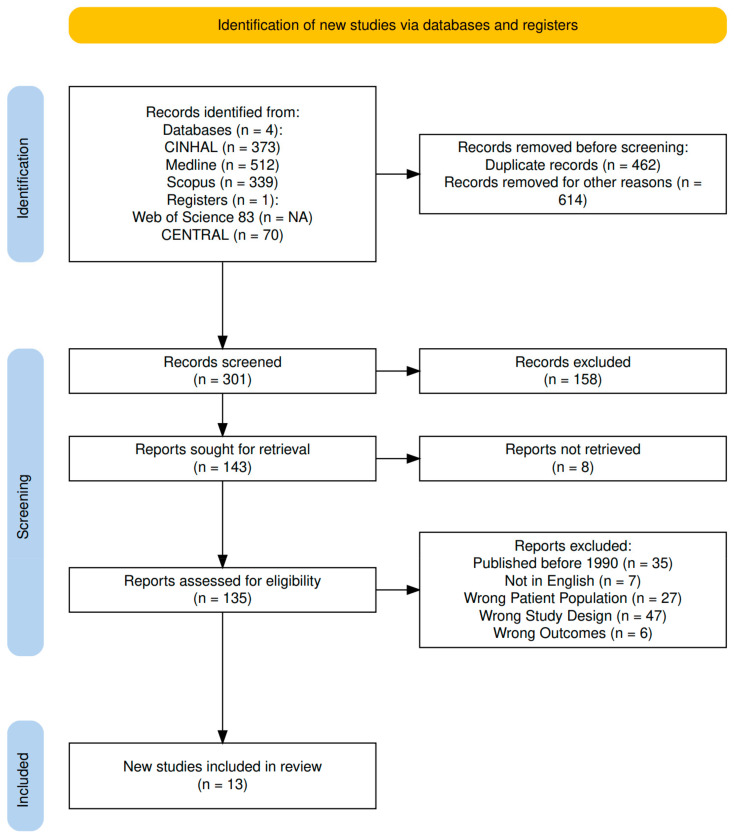
PRISMA flow chart.

**Table 1 healthcare-13-00531-t001:** Summary of the interventions of all included articles.

Study	Summary of Intervention
Abd El-Kafy et al., 2014 [19]	Compared three patient groups using TheraTogs, TheraTogs with sgrAFO, and physical therapy without orthotics through gait analysis by measuring speed, cadence, stride length, and hip and knee flexion angles in the mid-stance phase.
Carmick et al., 2012 [38]	Provided clinical observations of the effect of the SAFOs, HAFOs, and SMOs on the subtalar alignment through gait analysis focusing on range of motion, tibial torsion, ankle pronation, muscle strength, Gross Motor Function Measure, Pediatric Evaluation of Disability Inventory, and spatio-temporal parameters.
Davoudi et al., 2022 [33]	Compared three patient groups including barefoot, high-top shoes, and high-top shoes with TWO using gait analysis to measure center of pressure displacement in the anterior–posterior and medial–lateral axes.
Degelaen et al., 2016 [34]	Compared two patient groups with and without a 3D supporting garment using gait analysis to measure tridimensional trunk motion, trunk–thigh coordination, and interjoint coordination.
Flanagan, et al., 2009 [35]	A 12-week intervention using an individualized TheraTog garment system was designed. Using motion analysis, kinematic data including gait analysis, standing posture, gross motor skills test, and a biomechanical lower extremity assessment were collected.
Kim et al., 2022 [37]	Provided a custom-made tibia counter rotator (TCR) system to wear while sleeping, in combination with a gait plate (GP). The tibial transmalleolar angle (TMA) was evaluated through using a gravity goniometer.
Looper et al., 2012 [39]	Compared the effectiveness of SMOs and off-the-shelf FO through collecting data on calcaneal eversion and tibial torsion measurements. Three groups including barefoot, shoes with foot orthoses, and shoes with SMOs were compared by measuring tibial torsion, calcaneal eversion, and spatio-temporal parameters through gait analysis.
Mouri et al., 2019 [41]	Children were evaluated in MWI conditions using gait analysis to measure hip internal/external rotation, thigh–foot angle, femoral tibial angle, and metatarsus adductus index compared to a control population.
Munuera et al., 2010 [13]	Compared the effectiveness of three groups including unshod (AG1), shod without out-toeing wedge (AG2), and shod with out-toeing wedge (AG3) by measuring the angle of gait (FPA).
Parian et al., 2024 [42]	Two groups of children were included: gait plane insole and lateral sole wedge. FPA was measured barefoot before intervention and after orthotic use.
Richards et al., 2012 [20]	Two interventions lasting 6 weeks: 1 with TheraTog use and 1 with TC. Measured FPA, maximum knee extension, and hip rotation all in stance phase.
Rennie et al., 2000 [36]	Gait analysis was conducted using a Lycra garment and the root mean square error (RMSE) value was measured, specifically including transversal changes at the pelvis. Higher RMSE values expressed worsened stability.
Selby-Silverstein et al., 2001 [40]	Compared gait parameters in children with Down syndrome to children without disabilities by giving each group foot orthosis. Transverse plane foot angles were measured.

**Table 2 healthcare-13-00531-t002:** Summary of clinical studies on compression garments for gait correction in neuromuscular conditions.

Study	Type of Study and Level of Evidence	Number of Patients and Age	Orthotic Type	Outcomes
Abd El-Kafy et al., 2014 [19]	Three-armed randomized control trial (Level 1)	57 children with spastic diplegic cerebral palsy; ages 6–8 years.	TheraTogs AFOs	The TheraTogs with sgrAFOs group showed significant differences pre- and post-treatment compared to other groups in changes in hip and knee flexion angles in the mid-stance phase, with a 17.47° difference at right hip and 24.42° difference at right knee (*p* < 0.05).
Davoudi et al., 2022 [33]	Quasi-experimental study (Level 2)	20 children with spastic diplegic cerebral children with an in-toeing gait; average age of 6.8 ± 0.5 years.	Twister wrap orthoses	The combination of high-top shoes with TWO showed significant decreases in the center of pressure displacement in the medial–lateral direction of 28.8° (*p* < 0.001). Both orthotic interventions showed significant improvement in standing balance (*p* < 0.001).
Degelaen et al., 2016 [34]	Quasi-experimental study (Level 2)	15 children with bilateral spastic cerebral palsy; ages 4–10 years.	3D supporting compression garment	Significant changes seen in the coordination between trunk and lower limbs, step velocity, and cadence with garment use. Hip–knee and knee–ankle interjoint coordination improved during the stance phase (*p* < 0.05).
Flanagan, et al., 2009 [35]	Quasi-experimental study (Level 2)	5 children with diplegic cerebral palsy; ages 7–13 years.	TheraTogs	Increased peak hip extension at terminal stance and improved pelvic alignment in the sagittal plane with TheraTog use. Significant improvement seen in gross motor skills, Bruininks–Oseretsky Test of Motor Proficiency score improved from 22.4 to 35.2 with garment on (*p* < 0.05). Percent change in balance was increased positively by 76% at 4 months after intervention.
Rennie et al., 2000 [36]	Quasi-experimental study (Level 2)	7 children with cerebral palsy (CP) and 1 child with Duchenne muscular dystrophy (DMD); average age of 8.13 years (range from 5–11 years).	Lycra garment	5/8 children had decreased RMSE values at the pelvic level, improving proximal stability (non-significant, likely secondary to small sample size). No significant changes in mobility.

**Table 3 healthcare-13-00531-t003:** Summary of clinical studies on rotation systems for gait correction in neuromuscular conditions.

Study	Type of Study and Level of Evidence	Number of Patients and Age	Orthotic Type	Outcomes
Kim et al., 2022 [37]	Quasi-experimental study (Level 2)	124 Japanese pediatric patients with internal tibial torsion; age range of 3–15 years.	Tibia counter rotator	At 1 year post-treatment using TCR, tibial transmalleolar angle (TMA) was improved 5° in right leg (*p* < 0.01) and 0° in left leg (*p* < 0.01).
Richards et al., 2012 [20]	Case study (Level 4)	2-year-old child with L4 spina bifida with bilateral in-toeing.	TheraTogs Twister cables	TheraTog: Increased hip rotation of 23° left, 25° right, compared to baseline at 4° left, 11° right. Improved FPA on right at 30° and left at 13°. Compared to TheraTogs, TC had lower improvements with hip external rotation at −8° left, −9° right. TC had improved overall FPA.

**Table 4 healthcare-13-00531-t004:** Summary of clinical studies on ankle and foot orthotics for gait correction in neuromuscular conditions.

Study	Type of Study and Level of Evidence	Number of Patients and Age	Orthotic Type	Outcomes
Carmick et al., 2012 [38]	Case series (Level 4)	4 children with cerebral palsy; age range of 3.5–15 years.	SAFO SMO	Each case study showcased that a neutral subtalar joint position contributes to positive outcomes. If neutral position is not achieved by the orthotic, the hip, knee, and ankle alignment may be negatively impacted, leading to gait abnormalities and problems with ambulation.
Looper et al., 2012 [39]	Quasi-experimental study (Level 2)	6 children with Down syndrome; age range of 4–7 years.	SMO FO	Calcaneal eversion, navicular drop, and tibial torsion did not show significant correlations with gait parameters (*p* > 0.05).
Selby-Silverstein et al., 2001 [40]	Quasi-experimental study (Level 2)	16 children with Down syndrome; age range of 3–6 years.	FO	Transverse plane foot angle decreased external rotation by 7°, causing more internal rotation with foot orthoses (*p* < 0.001).

**Table 5 healthcare-13-00531-t005:** Summary of clinical studies on insoles and wedges for gait correction in neuromuscular conditions.

Study	Type of Study and Level of Evidence	Number of Patients and Age	Orthotic Type	Outcomes
Mouri et al., 2019 [41]	Quasi-experimental study (Level 2)	51 children with in-toeing gait symptom; average age of 5 years (range from 3–8 years).	MWI	In in-toeing group: significant increase seen in bilateral sum of hip internal rotation (136 ± 17°, *p* = 0.007). Significant decrease seen in bilateral sum of thigh–foot angle (−27 ± 21°, *p* < 0.001). Maximum foot range of motion was significantly increased (8.2 ± 3.0°, *p* = 0.002).
Munuera et al., 2010 [13]	Quasi-experimental study (Level 2)	48 children with an in-toed gait; age range of 3–14 years.	Out-toeing wedge	Comparison between groups: AG1 to AG2 by 1.66 degrees. AG1 to AG3 by 5.30 degrees. AG2 to AG3 by 3.60 degrees. Out-toeing wedges with shoes improved the angle of gait, improving gait pattern (*p* < 0.05).
Parian et al., 2024 [42]	Randomized control trial (Level 1)	11 children with in-toeing gait due to excessive femoral anteversion; age range of 7–10 years.	Gait plane insoleLateral sole wedge	Significant external improvement was seen in FPA compared to barefoot (*p* = 0.039). Gait plate insole with 3.51º change. Lateral sole wedge with 9.96 º change. FPA was improved significantly in the lateral sole wedge group (*p* = 0.013).

**Table 6 healthcare-13-00531-t006:** Changes in Hip and Knee Rotation in the transverse plane of different orthotic categories (mean ± SD) [20,35,37].

Orthotic Category	Level of Lower Extremity	Combined Mean ± SD (°)
Compression garments	Hip	19.73 ± 1.57
	Knee	NA
Rotational systems	Hip	8.50 ± 0.71
	Knee	24.13 ± 8.49

NA: the value cannot be computed.

**Table 7 healthcare-13-00531-t007:** Comparing FPA improvement of different orthotic categories (mean ± SD) [13,20,33,40,41,42].

Orthotic Category	Combined Mean ± SD (°)
Compression garments	4.86 ± 2.38
Rotational systems	19 ± 26.87
Foot orthotics	0 ± 9
Insoles and wedges	13.95 ± 16.95

## Data Availability

Data is contained within the article. Further data inquiry is available on request from the authors.

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
