# Peer review of "An Evaluation of Orthotics on In-Toeing or Out-Toeing Gait"

_healthcare, 2025, doi:10.3390/healthcare13050531_

Round 1
Reviewer 1 Report (Previous Reviewer 1)
Comments and Suggestions for Authors
This is a study that claims to be a systematic review, but it did not follow the PRISMA recommendations, although it said it did.
- There is no mention of the publication of the review protocol in PROSPERO.
- The inclusion and exclusion criteria do not take into account the acronym PICO, which does not have its components mentioned.
- They do not mention when the last search for articles occurred or the study design that the authors intended to find to be eligible for this review.
- According to Cochrane, the most important thing to assess the certainty of evidence in RCTs is the use of Cochrane's RoB tool and the use of the GRADE approach, which was not done in this review.
This study focuses more on clinical aspects of the studies than on the critical analysis of the literature, which is not a systematic review. This study presents more elements of a narrative or literature review.
In view of the above, the current version of the manuscript is no longer characterized as a systematic review, looking more like a literature review. Therefore, I reject the current version, and suggest to the authors to change the study to a narrative or literature review, since this study does not present the methodological robustness and method of a systematic review.
Author Response
Comments 1: There is no mention of the publication of the review protocol in PROSPERO.
Response 1: Thank you for your comment. This review was not registered in PROSPERO. By recommendation of our medical librarian at the Medical College of Wisconsin, we had opted to submit our protocol to OSF. The study protocol on Open Science Framework (osf.io) at [https://osf.io/ecqm2/] on [4/22/2024]. We have changed this manuscript from a systematic literature to a literature review, which does not require registration in PROSPERO.
Comments 2: The inclusion and exclusion criteria do not take into account the acronym PICO, which does not have its components mentioned.
Response 2: Thank you for your comment. We have now taken into account the PICO acronym. The changes can be found on page 3, line 109.
Comments 3: According to Cochrane, the most important thing to assess the certainty of evidence in RCTs is the use of Cochrane's RoB tool and the use of the GRADE approach, which was not done in this review.
Response 3: Thank you for your comment. Upon recommendation, we have transitioned this manuscript from a systematic literature review to a literature review, which does not require the use of a GRADE approach.
Comments 4: This study focuses more on clinical aspects of the studies than on the critical analysis of the literature, which is not a systematic review. This study presents more elements of a narrative or literature review.
Response 4: Thank you for your comment. We have transitioned this study as a literature review.
Comments 5: In view of the above, the current version of the manuscript is no longer characterized as a systematic review, looking more like a literature review. Therefore, I reject the current version, and suggest to the authors to change the study to a narrative or literature review, since this study does not present the methodological robustness and method of a systematic review.
Response 5: Thank you for your suggestion. We have transitioned this study as a literature review.
Reviewer 2 Report (Previous Reviewer 2)
Comments and Suggestions for Authors
This study aims to analyze Orthotics on In-toeing or Out-toeing Gait. The review seems very interesting and could be very useful for conservative treatment, so little in depth compared to surgical. The work is written in a satisfactory way. Congratulations. However, it would need some corrections that I recommend below:
Abstract: The results in the abstract should be enriched with statistical information to give a clearer message. The conclusions are too long.
The introduction is really too long. Please shorten and summarize it.
The table needs to be improved. It lacks information and the "Summary of study" needs to be more schematic. I would add some columns regarding: type of insoles, follow up, possible pain scales and scores.
The introduction should also include some information about insoles in general, their use in various pathologies, citing for example these recent articles:
- Chiou-Tan FY, Bloodworth D. Approach to gait disorders and orthotic management in adult onset neuromuscular diseases. Muscle Nerve. 2024 Aug 6. doi: 10.1002/mus.28208.
- Colò G, Leigheb M, Surace MF, et al. The efficacy of shoes modification and orthotics in hallux valgus deformity: a comprehensive review of literature. Musculoskelet Surg. 2024 Dec;108(4):395-402. doi: 10.1007/s12306-024-00839-9.
All evaluation parameters need a citation. Please provide.
line 399 - listing the levels of evidence is repetitive, please eliminate and make more summary conclusions.
References appear sparse for a systematic review. Please add at least 20 more recent citations.
The rest of article seems to flow fluently and I congratulate the authors.
Author Response
Comments 1: Abstract: The results in the abstract should be enriched with statistical information to give a clearer message. The conclusions are too long.
Response 1: Thank you for your comment. We agree that the abstract can be improved. You can find the improved results and shortened conclusion on pages 1 on line 19 and line 25, respectively.
Comments 2: The introduction is really too long. Please shorten and summarize it.
Response 2: We appreciate your comment. We have shortened the introduction which can be found on lines 35-95.
Comments 3: The table needs to be improved. It lacks information and the "Summary of study" needs to be more schematic. I would add some columns regarding: type of insoles, follow up, possible pain scales and scores.
Response 3: Thank you for your comment. We have added a “Summary of Study” table which is labeled as Table 1 and can be found on page X. For Tables 2-5, we have added a column on “Orthotic Type” as requested. Unfortunately, 2/13 articles discuss follow-up and limited studies discuss pain scores. For this reason, it is difficult to include columns on this criteria.
Comments 4: The introduction should also include some information about insoles in general, their use in various pathologies, citing for example these recent articles:
- Chiou-Tan FY, Bloodworth D. Approach to gait disorders and orthotic management in adult onset neuromuscular diseases. Muscle Nerve. 2024 Aug 6. doi: 10.1002/mus.28208.
- Colò G, Leigheb M, Surace MF, et al. The efficacy of shoes modification and orthotics in hallux valgus deformity: a comprehensive review of literature. Musculoskelet Surg. 2024 Dec;108(4):395-402. doi: 10.1007/s12306-024-00839-9.
Response 4: Thank you for the suggestion of these articles. We have included these articles into the manuscript. This can found on starting on line 83.
Comments 5: All evaluation parameters need a citation. Please provide.
Response 5: Thank you for your comment. We have included citations for the evaluation parameters that are discussed in the introduction and the methods section.
Comments 6: line 399 - listing the levels of evidence is repetitive, please eliminate and make more summary conclusions.
Response 6 Thank you for your comment. The conclusion has been edited with a summary conclusion statement on the level of evidence and quality of study.
Comments 7: References appear sparse for a systematic review. Please add at least 20 more recent citations.
Response 7: Thank you for this comment. We have transitioned this study from a systematic review to a literature review. We have added more articles and currently have 64 citations in the manuscript. It is difficult to find more articles given the nature of our topic and limited studies discussing affects in the transverse plane.
Reviewer 3 Report (New Reviewer)
Comments and Suggestions for Authors
Dear authors, the article "Evaluation of Orthotics on In-toeing or Out-toeing Gait: A Systematic Review" is very interesting. The study addresses a clinically relevant topic: the effectiveness of orthotic devices for managing in-toeing and out-toeing gait in children with neuromuscular disorders. However, there are some points of the paper that could be improved.
Material and methods.
1. Some statistical results lack clarity regarding variability (e.g., standard deviation) in reported changes across different orthotic groups. Ensure all numerical data in the results section include measures of dispersion (e.g., standard deviation or confidence intervals) for better interpretability.
Discussion
2. While the results are well-structured, the discussion section could further address the long-term implications of orthotic use and compliance challenges in pediatric populations. You can include more detailed insights into the clinical barriers to orthotic compliance (e.g., discomfort, usability, parental adherence).
3. The limitations are addressed, but the potential impact of publication bias is not sufficiently explored. Expand on the implications of publication bias on the study conclusions and suggest ways future research might address this limitation.
There are also some syntactic and orthographic mistakes,
Line 29.
"While AFOs and insoles or wedges present mixed evidence, achieving effective stabilization is crucial for AFOs to deliver benefits in the transverse plane."
Suggestion
"While AFOs and insoles or wedges present mixed evidence, achieving effective stabilization is crucial for AFOs to deliver measurable benefits in the transverse plane."
Line 31
"Conversely, foot orthotics (FOs) may be more appropriate for managing mild gait abnormalities."
Suggestion
"Foot orthotics (FOs), conversely, may be more appropriate for managing mild gait abnormalities."
Line 49
"Having femoral anteversion describes a deviated angle between the femoral neck and the femoral shaft due to forward torsion of the neck of the femur."
Suggestion
"Femoral anteversion refers to a deviated angle between the femoral neck and the femoral shaft caused by forward torsion of the femoral neck."
Author Response
Comments 1: Some statistical results lack clarity regarding variability (e.g., standard deviation) in reported changes across different orthotic groups. Ensure all numerical data in the results section include measures of dispersion (e.g., standard deviation or confidence intervals) for better interpretability.
Response 1: We understand your request to provide the standard deviation (SD) or confidence interval for each reported change. However, there are two combined SDs that cannot be directly calculated using the formula in Higgins et al, 2019, as we don’t have the SD information for the individual studies that are combined. We treated the sample means from the individual studies as our data and computed the SD based on it. This SD would mainly reflect between-study variation, which is different from the combined SD that accounts for both within-study and between-study variations. While not a perfect substitute, this measure can provide insight into the overall variability.
Comments 2: While the results are well-structured, the discussion section could further address the long-term implications of orthotic use and compliance challenges in pediatric populations. You can include more detailed insights into the clinical barriers to orthotic compliance (e.g., discomfort, usability, parental adherence).
Response 2: Thank you for your comments and suggestion. We have improved our discussion with more information regarding compliance challenges, along with adding additional references to support our findings. These improvements change be found on page 17, starting on line 376.
Comments 3: The limitations are addressed, but the potential impact of publication bias is not sufficiently explored. Expand on the implications of publication bias on the study conclusions and suggest ways future research might address this limitation.
Response 3: Thank you for this comment. We have expanded on the implication of publication bias in this study. These improvements change be found on page 18, starting on line 398.
Comments 4:
Line 29.
"While AFOs and insoles or wedges present mixed evidence, achieving effective stabilization is crucial for AFOs to deliver benefits in the transverse plane."
Suggestion
"While AFOs and insoles or wedges present mixed evidence, achieving effective stabilization is crucial for AFOs to deliver measurable benefits in the transverse plane."
Response 4: Thank you for your suggestion. We agree with you recommendation and appreciate the efforts you took to improve the clarity of our study.
Comments 5:
Line 31
"Conversely, foot orthotics (FOs) may be more appropriate for managing mild gait abnormalities."
Suggestion
"Foot orthotics (FOs), conversely, may be more appropriate for managing mild gait abnormalities."
Response 5: Thank you for your suggestion. We agree with you recommendation and appreciate the efforts you took to improve the clarity of our study.
Comments 6:
Line 49
"Having femoral anteversion describes a deviated angle between the femoral neck and the femoral shaft due to forward torsion of the neck of the femur."
Suggestion
"Femoral anteversion refers to a deviated angle between the femoral neck and the femoral shaft caused by forward torsion of the femoral neck."
Response 5: Thank you for your suggestion. We agree with you recommendation and appreciate the efforts you took to improve the clarity of our study.
This manuscript is a resubmission of an earlier submission. The following is a list of the peer review reports and author responses from that submission.
Round 1
Reviewer 1 Report
Comments and Suggestions for Authors
Thank you for the opportunity to review this article, which addresses an important topic in the area of ​​motor control in pediatrics. Although the article is interesting, it has many weaknesses and discrepancies in a systematic review. Below are my considerations about the article.
Abstract:
- Since this is a systematic review, it was not clear which instruments the authors used to evaluate the studies. Please include them.
- In addition, in the conclusion of the abstract, the authors do not mention the certainty of the evidence. What is the quality of the studies analyzed? Can we trust this evidence? This needs to be better described in the abstract, including in the objective of the study.
Introduction:
- The introduction is quite extensive and goes into great detail about what walking with the toes out and in can cause. I believe the authors could reduce these paragraphs more, making the text shorter and more focused so that the reader does not feel tired when reading the article.
- I noticed that the objective of the study did not include an assessment of the quality/certainty of the evidence. A systematic review differs from other types of reviews in that it is able to judge the quality/certainty of the evidence in the studies. If this is not the objective of this study, I suggest changing it to a literature review or integrative review.
Methods:
- Was the protocol for this review not registered in PROSPERO?
- The authors need to standardize the databases in which the searches were carried out. The databases searched and mentioned in the abstract and methods are divergent.
- What were the inclusion and exclusion criteria for this review? This needs to be clear in the methods.
- How old were the children? What was the disease of the children? What was the type/design of the study? What type of orthosis? What outcomes were analyzed? All of this information needs to be included in the review methods.
- How were the data analyzed?
- Was there no critical analysis of the literature on the topic? This would invalidate the systematic review.
Results:
- It is not possible to conduct a systematic review with several types of studies.
- There is no mention of data analysis in the methods, however, in the results section we have three figures with data. Therefore, it is difficult to interpret how this data was collected.
Discussion:
- The discussion is long and presents what the studies have shown for each type of orthosis, which is typical of a literature review.
- The current discussion does not fit into a discussion of a systematic review, especially in terms of what PRISMA recommends, as the authors said that this review was conducted, which is not true.
Conclusion:
- The conclusion made no mention of the quality/certainty of the evidence, nor did it mention what would be important in terms of recommendations for future studies; the current conclusion is generalist and not very specific.
Author Response
Thank you very much for taking the time to review this manuscript. Please find the detailed responses below.
Comments 1: Since this is a systematic review, it was not clear which instruments the authors used to evaluate the studies. Please include them.
Response 1: Thank you for pointing this out. We have completed the necessary risk of bias assessments for the respective articles. This description of the instruments used to evaluate the studies is located on Page 4 under 2.6 Quality Assessment, starting at line 165.
Comments 2: In addition, in the conclusion of the abstract, the authors do not mention the certainty of the evidence. What is the quality of the studies analyzed? Can we trust this evidence? This needs to be better described in the abstract, including in the objective of the study.
Response 2: Thank you for your comment. A risk of bias assessment was completed. The abstract has been improved with description of the quality of the studies. An additional objective was added, which can be found on Page 4 at the end of the introduction. Additionally, you can find the risk of bias discussed in the methods (section 2.6 Quality assessment, line 165) and results section (line 225). Tables for the risk of bias are included in the Appendix B section, starting at line 443.
Comments 3: The introduction is quite extensive and goes into great detail about what walking with the toes out and in can cause. I believe the authors could reduce these paragraphs more, making the text shorter and more focused so that the reader does not feel tired when reading the article.
Response 3: We agree with your suggestion. You can find a shortened introduction on pages 1-4, beginning at line 38. Specific details regarding biomechanics have been reduced significantly.
Comments 4: I noticed that the objective of the study did not include an assessment of the quality/certainty of the evidence. A systematic review differs from other types of reviews in that it is able to judge the quality/certainty of the evidence in the studies. If this is not the objective of this study, I suggest changing it to a literature review or integrative review.
Response 4: Thank you for your suggestion. A quality assessment of the included studies is now completed. This description of the instruments used to evaluate the studies is located on Page 4 under 2.6 Quality Assessment, starting at line 165.
Comments 5: Was the protocol for this review not registered in PROSPERO?
Response 5: This review was not registered in PROSPERO. By recommendation of our medical librarian at the Medical College of Wisconsin, we had opted to submit our protocol to OSF. The study protocol on Open Science Framework (osf.io) at [https://osf.io/ecqm2/] on [4/22/2024].
Comments 6: The authors need to standardize the databases in which the searches were carried out. The databases searched and mentioned in the abstract and methods are divergent.
Response 6: Thanks for your comment. The databases searched should now be accurately described in both the abstract and the methods section in the Search Strategy section, beginning at line 115.
Comments 7: What were the inclusion and exclusion criteria for this review? This needs to be clear in the methods.
Response 7: Thank you for your suggestion in making the inclusion and exclusion clearer for the reader. These changes can be found in the methods section on page 4 under 2.3 Eligibility Criteria, starting at line 124.
Comments 8: How old were the children? What was the disease of the children? What was the type/design of the study? What type of orthosis? What outcomes were analyzed? All of this information needs to be included in the review methods.
Response 8: Thank you for your comment. Demographic information has been added to the methods, the majority of which can be found on page 5, section 2.7 Outcome Measures at line 188.
Comments 9: How were the data analyzed?
Response 9: Thank you for the comment. Details on the data analysis have been added to the methods section, under 2.8 Statistical Analysis at line 200. The differences of the mean FPA and mean joint kinematics from the included studies were compared for each orthotic category. Statistical analyses for combing means and standard deviation were performed using R version 4.3.2.
Comments 10: Was there no critical analysis of the literature on the topic? This would invalidate the systematic review.
Response 10: Thank you for your comment. Critical analysis of the literature has been added in the discussion section along with quality assessment, starting at line 270.
Comments 11: It is not possible to conduct a systematic review with several types of studies.
Response 11: Thank you for your comment. The gathering of the articles in the databases was guided by our medical librarians at the Medical College of Wisconsin. The topic of kinematic changes in the transverse plane is limited in the literature, making it difficult to only include one type study. Broadening the category of the studies has helped us provide more context on this topic in order to enhance our comparison of the four main categories of orthotics we established.
Comments 12: There is no mention of data analysis in the methods, however, in the results section we have three figures with data. Therefore, it is difficult to interpret how this data was collected.
Response 12: Thank you for your comment. Details on the data analysis have been added to the methods section, under 2.5 Data Collection at line 153.
Comments 13: The discussion is long and presents what the studies have shown for each type of orthosis, which is typical of a literature review. The current discussion does not fit into a discussion of a systematic review, especially in terms of what PRISMA recommends, as the authors said that this review was conducted, which is not true.
Response 13: We agree with your comment. The discussion has been edited and transformed into one that fits more into a systematic literature. By recommendation of other reviewers, we added more recent citations to contextualize the findings of our included articles.
Comments 14: The conclusion made no mention of the quality/certainty of the evidence, nor did it mention what would be important in terms of recommendations for future studies; the current conclusion is generalist and not very specific.
Response 14: Thank you for your comment. The conclusion has been edited with changes regarding the quality of evidence and recommendations for future studies at line 391.
Reviewer 2 Report
Comments and Suggestions for Authors
This study aims to analyze Orthotics on In-toeing or Out-toeing Gait. The study appears very interesting and well written. Congratulations to all the authors. However, it requires some corrections which I recommend here below:
The introduction is really too long. Please shorten and summarize.
The references in the introduction should be improved and enriched. Nowadays, plantar orthoses are also the subject of debate for other anomalies. Make a small initial overview citing for example some recent articles such as:
- Colò G, Leigheb M, Surace MF, Fusini F. The efficacy of shoes modification and orthotics in hallux valgus deformity: a comprehensive review of literature. Musculoskelet Surg. 2024 Jun 26. doi: 10.1007/s12306-024-00839-9.
- Cooper S, Hanning J, Hegarty C, Generalis C, Smith A, Hall T, Starbuck C, Kaux JF, Schwartz C, Buckley C. Effects of a range of 6 prefabricated orthotic insole designs on plantar pressure in a healthy population: A randomized, open-label crossover investigation. Prosthet Orthot Int. 2024 Aug 1;48(4):474-480. doi: 10.1097/PXR.0000000000000292.
A subsection of the statistical analysis is missing. Please provide.
Even discussion appears too long. Please shorten and summarize.
References appear very sparse. Please add at least 10-20 more recent citations.
The conclusions should be enriched with statistical information to give a clearer message.
The rest of article seems to flow fluently and I congratulate the authors.
Author Response
Thank you very much for taking the time to review this manuscript. Please find the detailed responses below.
Comments 1: The introduction is really too long. Please shorten and summarize.
Response 1: Thank you for your comment. We agree that the introduction can be improved. You can find a shortened introduction on pages 1-3. Specific details regarding biomechanics have been reduced significantly.
Comments 2: The references in the introduction should be improved and enriched. Nowadays, plantar orthoses are also the subject of debate for other anomalies. Make a small initial overview citing for example some recent articles such as:
- Colò G, Leigheb M, Surace MF, Fusini F. The efficacy of shoes modification and orthotics in hallux valgus deformity: a comprehensive review of literature. Musculoskelet Surg. 2024 Jun 26. doi: 10.1007/s12306-024-00839-9.
- Cooper S, Hanning J, Hegarty C, Generalis C, Smith A, Hall T, Starbuck C, Kaux JF, Schwartz C, Buckley C. Effects of a range of 6 prefabricated orthotic insole designs on plantar pressure in a healthy population: A randomized, open-label crossover investigation. Prosthet Orthot Int. 2024 Aug 1;48(4):474-480. doi: 10.1097/PXR.0000000000000292.
Response 2: We appreciate your comment. Thank you for the recommendation of these recent articles. We have included these articles in the introduction, which can be found on page 3 at line 98.
Comments 3: A subsection of the statistical analysis is missing. Please provide
Response 3: Thank you for your comment. Details on the data analysis have been added to the methods section, under 2.8 Statistical Analysis at line 200. The differences of the mean FPA and mean joint kinematics from the included studies were compared for each orthotic category. Statistical analyses for combing means and standard deviation were performed using r version 4.3.2.
Comments 4: Even discussion appears too long. Please shorten and summarize.
Response 4: We agree with your comment. We have edited the discussion. With suggestions from other reviewers, the discussion has been re-formatted to account for quality assessment and accordance with PRISMA guidelines, starting at line 270.
Comments 5: References appear very sparse. Please add at least 10-20 more recent citations.
Response 5: Thank you for your comment. Along with your recommendation of 2 articles regarding plantar orthoses, we added 5 additional recent articles to contextualize findings in the discussion. The topic regarding changes in the transverse plane is limited to literature and limited for certain orthotics, especially those in the rotational system categories. Due to this, it is difficult to add additional articles that are relevant to the discussion.
Comments 6: The conclusions should be enriched with statistical information to give a clearer message.
Response 6 Thank you for your comment. The conclusion has been edited with information regarding quality assessment and level of evidence at line 391. Additionally, we have added information regarding recommendations for future studies.
Comments 7: The authors need to standardize the databases in which the searches were carried out. The databases searched and mentioned in the abstract and methods are divergent.
Response 7: Thank you for catching this mistake. The databases searched should now be accurately described in both the abstract and the methods section in the Search Strategy section, beginning at line 115.
Reviewer 3 Report
Comments and Suggestions for Authors
I would like to thank the editors for the possibility to evaluate this systematic review without meta-analysis “The Evaluation of Orthotics on In-toeing or Out-toeing Gait: A Systematic Review”.
The introduction is sufficiently explanatory and explains the main current findings, as well as the justification for the review and its objective. It uses updated bibliography.
PICO or similar questions are not indicated in systematic reviews. The criteria for inclusion and exclusion of articles should also be better specified. Perhaps the review should be limited to pediatric population only, since as the authors indicate “19 studies were based on pediatric patients and 2 studies included adult patient”.
In the search they indicate the key words, in addition to offering the search strings in the supplementary material. The search is performed in the main databases, although PROSPERO (https://www.crd.york.ac.uk/PROSPERO/) is missing. It is recommended that the review be registered in PROSPERO.
It is necessary to evaluate the articles found and selected by means of a quality assessment scale such as AMSTAR (amstar.ca/index.php) OR PEDRo (pedro.org.au/). In section 3.1, several levels of evidence are indicated, but there is no bibliographic reference, so it is not known how they have been classified.
The 21 articles selected include 3 systematic reviews that should not and cannot be included in a review. Then in Table 5 there are 4 Systematic Reviews and Meta-Analyses on the Effectiveness of Orthoses for Improving Rotational Deformities.
The discussion is correct and adheres to the main results obtained.
They should review the bibliography since it appears in different systems, thus citations 1, 2, 4, 10, 12, 21, 29 and 34 have the name of the journal in italics. In 36 they leave the ... in the authors. In general, they should revise the bibliography and adjust it to the norms.
Author Response
Thank you very much for taking the time to review this manuscript. Please find the detailed responses below.
Comments 1: PICO or similar questions are not indicated in systematic reviews. The criteria for inclusion and exclusion of articles should also be better specified. Perhaps the review should be limited to pediatric population only, since as the authors indicate “19 studies were based on pediatric patients and 2 studies included adult patient”.
Response 1: Thank you for your suggestion in making the inclusion and exclusion clearer for the reader. These changes can be found in the methods section on under 2.3 Eligibility Criteria at line 124. Additionally, we agree with your suggestion on focusing on the pediatric population. The 2 adult studies have been removed from the paper.
Comments 2: In the search they indicate the key words, in addition to offering the search strings in the supplementary material. The search is performed in the main databases, although PROSPERO (https://www.crd.york.ac.uk/PROSPERO/) is missing. It is recommended that the review be registered in PROSPERO.
Response 2: Thank you for your comment. This review was not registered in PROSPERO. By recommendation of our medical librarian at the Medical College of Wisconsin, we had opted to submit our protocol to OSF. The study protocol on Open Science Framework (osf.io) at [https://osf.io/ecqm2/] on [4/22/2024].
Comments 3: It is necessary to evaluate the articles found and selected by means of a quality assessment scale such as AMSTAR (amstar.ca/index.php) OR PEDRo (pedro.org.au/). In section 3.1, several levels of evidence are indicated, but there is no bibliographic reference, so it is not known how they have been classified.
Response 3: We agree with this comment. We have completed the necessary risk of bias assessments for the respective articles. This description of the instruments used to evaluate the studies is located on Page 4 under 2.6 Quality Assessment at line 165. Additionally, the levels of evidence are now cited under the same section.
Comments 4: The 21 articles selected include 3 systematic reviews that should not and cannot be included in a review. Then in Table 5 there are 4 Systematic Reviews and Meta-Analyses on the Effectiveness of Orthoses for Improving Rotational Deformities.
Response 4: Thank you for your suggestion. We have removed old Table 5 containing information on the systematic review articles and the 3 systematic reviews discussed in the manuscript. The new Table 5 contains data from the included clinical studies on the rotational changes for each respective orthotic category.
Comments 5: They should review the bibliography since it appears in different systems, thus citations 1, 2, 4, 10, 12, 21, 29 and 34 have the name of the journal in italics. In 36 they leave the ... in the authors. In general, they should revise the bibliography and adjust it to the norms.
Response 5: Thank you for your comment. We have incorporated a citation tool manager “Zotero” which has helped us improve the errors in the bibliography.